# Validity and Reliability of the Korean Version of the Holistic Nursing Competence Scale

**DOI:** 10.3390/ijerph19127244

**Published:** 2022-06-13

**Authors:** Kawoun Seo, Taejeong Jang, Taehui Kim

**Affiliations:** 1Department of Science of Nursing, Joongbu University, 201, Deahak-ro, Chubu-myeon, Geumsan-gun 32713, Korea; kwseo@joongbu.ac.kr; 2College of Nursing, Woosuk University, 443 Samnye-ro, Samnye-eup, Wanju-gun 55338, Korea

**Keywords:** competence, Holistic Nursing Competence Scale, methodological study, nursing care

## Abstract

This methodological study aimed to verify the validity and reliability of the Korean version of the Holistic Nursing Competence Scale (HNCS), which comprises five dimensions and 36 items. The English version of the HNCS was forward and backward translated and administered to 251 participants with more than a year of work experience in a general hospital. Data were analyzed using SPSS WIN 24.0(Chicago, IL, USA), and AMOS program was used for confirmatory factor analysis. Additionally, the “Task Performance Evaluation Instrument for Clinical Nurses” was used for concurrent validity. Reliability assessed using Cronbach’s α was 0.969. Convergent, discriminant, and concurrent validity were good. Average variance extracted and construct reliability ranged from 0.845 to 0.932 and 0.980 to 0.987, respectively. The model was suitable with the chi-square value being 1216.563 (df = 584, *p* < 0.001), and Q value being less than three. Goodness-of-fit index, root mean square residual, and root mean square error of approximation were 0.784, 0.066, and 0.066, respectively. Moreover, comparative fit index, Tucker–Lewis index, and incremental fit index were 0.913, 0.906, and 0.913, respectively. Thus, this study verified the validity and reliability of the Korean version of the HNCS. Our findings suggest that the scale is helpful in measuring and developing the holistic nursing competence of clinical nurses.

## 1. Introduction

Competence is the ability to achieve desirable results by integrating both capability and performance. It includes the totality of knowledge, skills, and attitudes required to perform nursing, and refers to the characteristics of behavior that lead to the successful performance of one’s duties [1,2]. Nursing competence is defined as the “sets of knowledge, skills, traits, motives, and attitudes that are required for effective performance in a wide range of nursing jobs and various clinical settings” [3].

Holistic nursing emphasizes a multidimensional approach to providing nursing care; it involves providing comprehensive nursing to patients, including physical, psychological, sociological, and spiritual aspects [4]. “Holistic” means more than the sum of the parts, and holistic nursing considers sociological and psychological wellbeing as well as the treatment of disease [4]. Therefore, holistic nursing competence is the ability to provide effective and professional nursing care to patients with a professional attitude, values, knowledge, skills, and responsibility for nursing performance while approaching patients from a holistic perspective [5].

Nurses account for the largest proportion of the hospital workforce and are a major factor influencing patient outcomes such as complication, infection, and death [6]. However, due to nurse shortages and excessive workloads, nurses are unable to provide holistic nursing [1], something that is critical following the Fourth Industrial Revolution. The Fourth Industrial Revolution enables information and communication technology-based smart healthcare services, including robots and artificial intelligence (AI), which can provide medical-record-based nursing to patients [7]. Simple nursing tasks, such as record-keeping, taking vital signs of stable patients, and testing blood sugar, can be performed by robots or AI. This allows nurses to provide more patient-centered nursing care, which focuses on communication, shared understanding, and improved quality of nursing service and is increasingly required of nurses [7]. In this context, holistic nursing competence is essential for providing patient-centered care. 

In Korea, a family member is allowed to nurse an inpatient in the hospital, including fundamental nursing care such as defecation, excretion, meal assistance, and tracheal suction. However, increased emphasis on individual quality of life and other social changes have resulted in a decline in provision of family care. Many people hire caregivers to perform inpatient care, which can be costly. To help individuals with financial difficulties, the Korean government introduced the integrated nursing care service in 2013 and began to establish hospitals without family caregivers [8]. In this integrated nursing care service, all care is provided by nurses with no family caregivers. Additionally, the prevalence of respiratory infectious diseases such as influenza, a virus subtype H1N1 in 2008, Middle East respiratory syndrome (MERS) in 2015, and Coronavirus disease 2019 (COVID-19) in 2019 increased the number of patients in need of isolation treatment. Such treatment included a variety of nursing activities, ranging from simple care of isolated patients to intensive care, because of the variety of severe illnesses [9]. 

The isolation room is a space where family caregivers are not allowed to enter. Therefore, nurses must render all the care services, such as providing drinks and meals, moving the patient to the bathroom, and washing, which are typically provided by family caregivers. Moreover, isolation is likely to cause psychosocial problems such as depression and anxiety [10]. Thus, holistic nursing is important for such patients [9]. Holistic nursing competence is necessary to provide patient care from a holistic perspective, considering the social and environmental changes. 

Nurses in Korea are evaluated according to the clinical career ladder system, which evaluates their abilities based on their career and nursing competence. However, this system utilizes scales that are segmented by career to increase nurses’ competence. The concurrent validity was verified using the “Task Performance Evaluation Instrument for Clinical Nurses”, which was developed to evaluate the clinical work of both staff nurses and supervisors. Furthermore, this self-report tool is divided into four dimensions: knowledge, attitude, ethics, and performance. Although there are tools that can be utilized for job evaluation, it is difficult to evaluate the holistic nursing competence that is universally required for all nurses using these tools. 

Additionally, no scale exists that can measure holistic nursing competence with a small number of questions. Above all, it is important for nurses to evaluate their own competence and recognize their inadequacies [1].

Based on the above, this study aimed to verify the reliability and validity of the Holistic Nursing Competence Scale (HNCS) developed by Takase and Teraoka [5]. 

## 2. Materials and Methods

### 2.1. Study Design

This study used a methodological study design. The HNCS, developed by Takase and Teraoka, was translated into Korean, and the validity and reliability of the Korean version were verified [5].

### 2.2. Participants

The study participants were nurses conveniently sampled from four general hospitals who voluntarily participated. Inclusion criteria were (a) working for more than one year and (b) directly caring for patients. New nurses with less than a year of work experience were excluded because it was assumed they face more difficulties adapting to and performing their job [11,12]. Moreover, nurses working in departments that did not provide direct nursing care, such as outpatient clinics, departments of infection control and management, departments of healthcare quality management, and administrative departments, were excluded [5]. This was because the test items explored relationships, communication with patients, and education for patients and caregivers, which are not applicable to work in such departments. The recommended sample size for the confirmatory factor analysis (CFA) using AMOS was 200–400 [13]. Thus, data from 251 participants were used for analysis.

### 2.3. Measurements

#### 2.3.1. Holistic Nursing Competence Scale

The HNCS, developed by Takase and Teraoka [5], is used to measure the holistic nursing competence of nurses who directly care for patients. It is a seven-point Likert scale comprising 36 items. Tool scores range from 36 to 252 points; the higher the score, the higher the holistic nursing competence. The HNCS comprises five sub-areas—staff education and management, ethically oriented practice, general attitude, nursing care in a team, and professional development—with nine, nine, seven, seven, and four items, respectively. Takase and Teraoka [5] used the following Likert scale for a pilot test: 1 = not at all, 2 = seldom, 3 = occasionally, 4 = sometimes, 5 = frequently, 6 = nearly always, and 7 = always. After the pilot, the scale was modified: 1 = not competent at all, 2 = slightly competent, 3 = somewhat more competent, 4 = reasonably competent, 5 = almost fully competent, 6 = fully competent, and 7 = extremely competent. However, after discussing with the researchers and clinical nurses, the former Likert scale, being more familiar to nurses, was used in this study. Takase and Teraoka [5] performed an exploratory factor analysis among 331 nurses to verify the construct validity of the original scale. Its reliability was Cronbach’s α = 0.967. In this study, the reliability was Cronbach’s α = 0.969.

#### 2.3.2. Task Performance Evaluation Instrument for Clinical Nurses 

The Task Performance Evaluation Instrument for Clinical Nurses is a tool for evaluating the work performance ability of clinical nurses [14]. In most evaluation methods used in hospitals, supervising nurses evaluate staff nurses [14]. This tool was designed to address the problem of the one-sided evaluation method by superiors. In other words, it simultaneously considers nurses’ self-assessment and supervisors’ evaluation. It comprises four dimensions and 35 items: knowledge (8 items), attitude (13 items), performance ability (7 items), and ethics (7 items). According to several researchers, this tool contains the necessary components to assess holistic nursing care and nurses’ competence [15]. The items on it match the HNCS. This questionnaire employs a five-point Likert scale ranging from one to five with scores ranging from 35 to 175. The higher the score, the better the task performance. The reliability of the questionnaire was 0.918 at the time of development and 0.959 in this study.

### 2.4. Procedure

First, permission to use the scale was obtained from the original authors who developed the HNCS [5]. After this, the study was conducted in three stages: (1) translation and reverse translation, (2) content validation, and (3) evaluation of the psychometric properties. The validity was verified by determining the content, construct, and criterion validity, as well as the reliability of the scale. 

#### 2.4.1. Translation/Reverse Translation

The method of double translation suggested by Waltz et al. [16] was used for the translation process. For the primary translation, three professors of nursing departments who could speak both Korean and English translated the original scale into Korean. After discussion, the primary translation was completed. A bilingual expert who majored in nursing science translated it back to English. After that, the difference in meaning between the original scale and the reverse-translation scale was confirmed. It was found that there was no particular difference in meaning between the questions, and the Korean questionnaires were confirmed after discussion among researchers.

#### 2.4.2. Content Validity

The confirmed Korean questionnaire was evaluated for the appropriateness of the words used for understanding the questions by a chief nurse and five staff nurses. After that, it was modified based on discussion and debate among researchers, translators, and reverse-translators. The word “practice” in items 11, 12, and 34 was revised to “task”, the word “communication” in item 14 was revised to “talk”, and the passive sentence in item 35 was revised to be active. 

#### 2.4.3. Psychometric Properties

Construct and concurrent validity as well as internal consistency reliability were assessed to evaluate the psychometric properties of the HNCS. 

#### 2.4.4. Ethical Consideration

This study was approved by the Institutional Review Board of Joongbu University (IRB No. JIRB-2021050301-01) prior to study initiation. Data were collected in two ways from 31 May to 30 June 2021. First, the researcher visited the nursing departments of university hospitals and general hospitals; explained the research purpose, contents, methods, and so on; and asked for cooperation. Before conducting the survey, participants were told the purpose, contents, and procedure of the study as well as the time required to complete the survey, whether it was possible to voluntarily participate or withdraw, the processing method and use of the research data, and the guarantee of confidentiality. Next, the nurses who expressed their intention to participate voluntarily signed the research consent form and completed the questionnaire. The collected data were coded and entered into the computer by the researcher after the survey was completed, and the questionnaire papers were then locked in a cabinet. For the second method of data collection, participants were recruited through snowball sampling, and the survey was administered to the nurses who consented to participate using an online survey form. The participants who completed the questionnaire and provided their phone numbers received a gift (coffee voucher).

### 2.5. Statistical Analysis

The collected data were analyzed using SPSS WIN 24.0 (Chicago, IL, USA), and AMOS program. The general characteristics of the participants were analyzed using descriptive statistics. The validity was confirmed through CFA. Factor analysis is a useful approach for assessing construct validity [16]. The model fit of the HNCS’ Korean version was verified using χ^2^, CMIN/df, goodness-of-fit index (GFI), root mean square residual (RMR), root mean square error of approximation (RMSEA), comparative fit index (CFI), Tucker–Lewis index (TLI), and incremental fit index (IFI). The convergent validity of the scale was confirmed by checking the standardized factor loading (SFL), critical ratio, construct reliability (CR), and average variance extracted (AVE). Discriminant validity was verified using the correlation coefficient and AVE value. Concurrent validity was verified using Pearson’s correlation with the Task Performance Evaluation Instrument for Clinical Nurses [14]. The reliability of the Korean version of the HNCS was confirmed using Cronbach’s α.

## 3. Results

### 3.1. General Characteristics

Table 1 shows the general characteristics of the participants. Data were analyzed for 251 of the 300 participants. The average age was 30.67 ± 5.48 years, and 90.8% were female. The highest education level was a bachelor’s degree (84.5%). The average clinical experience was 6.51 ± 5.68 years, and nurses who had worked for less than one to five years were the most common. Of the 251 analyzed participants, 97.2% were staff nurses, 37.8% worked at intensive care units, and 62.2% worked at general wards. Factor 2 (5.311 ± 0.875) had the highest average item score by factor, and factor 1 (4.116 ± 1.064) had the lowest.

### 3.2. Item Analysis

The mean for each item was 3.93–5.69. The skewness ranged from −0.63 to 0.25 and the kurtosis ranged from −0.64 to 0.5. It was presented in Table 2. 

### 3.3. Content Validity 

Ten experts determined the content validity of the Korean version of the HNCS on a 4-point scale: 1 = not relevant, 2 = somewhat relevant, 3 = quite relevant, and 4 = highly relevant [17]. The content validity index was >0.8. 

### 3.4. Construct Validity 

#### 3.4.1. Assessing the Fit of the Model 

CFA was performed on five sub-areas of the original scale to verify the convergent, discriminant, and criterion validity of the HNCS. The results are shown in Table 3. The model was suitable since the chi-square value was 1216.563 (df = 584, *p* < 0.001) and the Q value (CMIN/df) was less than 3 [18].

GFI, RMR, and RMSEA were 0.784, 0.066, and 0.066, respectively. Furthermore, CFI, TLI, and IFI were 0.913, 0.906, and 0.913, respectively. The factor loads are shown in Figure 1. Therefore, structural equation model fit was accepted. 

#### 3.4.2. Convergent Validity

The convergent validity of the Korean version of the HNCS was confirmed by checking the SFL for each item, AVE, and CR. SFL ranged from 0.61–0.89 and satisfied the criterion because it was greater than 0.50. AVE value ranged from 0.845–0.932, which met the criterion of more than 0.50. CR value ranged from 0.980–0.987, which satisfied the criterion of more than 0.70. Table 4 presents the detailed results.

#### 3.4.3. Discriminant Validity

To check the discriminant validity, the AVE and correlation coefficient square of the five factors were compared (Table 5). The discriminant validity was verified as the correlation coefficient square for each factor was smaller than the AVE.

### 3.5. Concurrent Validity

Table 6 shows the criterion validity results. The relationship between the HNCS and Task Performance Evaluation Instrument for Clinical Nurses was analyzed to verify the criterion validity. The two scales showed a statistically significant positive correlation (r = 0.712, *p* < 0.01).

### 3.6. Reliability

Cronbach’s α of all 36 items of the HNCS Korean version was 0.969. Additionally, Cronbach’s α for each sub-area was 0.933 for staff education and management, 0.928 for ethically-oriented practice, 0.914 for general attitude, 0.919 for nursing care in a team, and 0.899 for professional development. 

## 4. Discussion

Holistic nursing competence is essential for nurses and does not change across time and environment. The HNCS was developed to easily and practically measure the ability of clinical nurses by reflecting on 10 attributes of nursing competence: personal characteristics, cognitive ability, orientation toward ethical/legal practice, engagement in professional development, collaboration with other healthcare professionals, providing teaching/coaching to patients and staff, demonstrating management skills, ensuring quality and safety in care, establishing interpersonal relationships with patients and nursing staff, and managing nursing care. The HNCS is divided into sections A and B. Section A is factor 3, measuring the general characteristics of nurses, such as critical thinking, self-reflection, and compassion. Section B is factors 1, 2, 4, and 5, evaluating the professional characteristics of nurses. 

This study was conducted to evaluate the appropriateness of the Korean version of the HNCS to easily measure the holistic nursing competence. As it is more appropriate to conduct confirmatory than exploratory factor analysis for a scale whose validity is confirmed [19], CFA was conducted. First, comparing the general characteristics with the original scale, the average age (Korean 30.67 ± 5.48 vs. original 29.79 ± 8.10), length of clinical experience (Korean 6.51 ± 5.98 vs. original 6.81 ± 7.45), and marital status (Korean: single 73.3% vs. original: single 74.9%) showed similar characteristics. The average scores for the factors were Korean 4.116 ± 1.064 vs. original 3.649 ± 1.079 for factor 1, Korean 5.311 ± 0.875 vs. original 4.713 ± 0.948 for factor 2, Korean 5.059 ± 0.920 vs. original 4.299 ± 0.784 for factor 3, Korean 5.274 ± 0.911 vs. original 4.536 ± 0.847 for factor 4, and Korean 4.935 ± 0.918 vs. original 4.228 ± 0.792 for factor 5. The average score in the Korean version was slightly higher, and this was attributed to a difference in education; 48.8% of the participants in the original scale had a bachelor’s degree or higher, while in the Korean version, 89.3% had a bachelor’s degree or higher.

Regarding model fit, the GFI was 0.784, which did not exceed 0.9 [20]. RMSEA was 0.066; as RMSEA < 0.05 is considered a close fit and 0.50 < 0.08 is considered a reasonable fit [21], RMSEA showed appropriate results. Incremental fit indices GFI, TLI, and IFI were 0.913, 0.906, and 0.913, respectively. These results were appropriate because they were above 0.9. Therefore, the model showed suitable fit for the Korean version of the HNCS. 

Regarding the convergent validity of the items, the SFL and AVE of all items satisfied the criterion of 0.50 or more, and the CR was also 0.70 or more. Therefore, the convergent validity was confirmed [13]. To confirm discriminant validity, the relationship between AVE and correlation coefficient square of the five factors were examined. The discriminant validity satisfied the criterion since AVE was larger than the correlation coefficient square. The internal reliability found in this study was Cronbach’s α = 0.969, which was similar to the reliability of the original scale (0.967). Cronbach’s αs of the original scale for factors 1, 2, 3, 4, and 5 were 0.934, 0.934, 0.868, 0.919, and 0.881, respectively. This was similar to the values of the Korean version (0.933, 0.928, 0.914, 0.919, and 0.899, respectively). Therefore, the reliability of the Korean version of the HNCS was high, similar to that of the original scale. Moreover, based on this, the model fit, validity, and reliability of the Korean version of the HNCS were verified.

Takase used the seven-point rating scale asking about frequency in the pilot test and competence in the field test. Frequency was used in this study. It is possible to respond objectively to questions about frequency; asking about competence involves high-level evaluation of one’s ability and, thus, involves subjective judgement. Future studies will require comparing “asking about frequency” and “asking about competence”.

Takase and Teraoka stated that there is a correlation between clinical experience and holistic nursing competence [5]. Nursing competence increases according to the order of year and work experience of nurses as professionals, and a difference in competence is seen depending on the field of nursing [22]. It also depends on the quality and quantity of clinical experience and individual efforts to improve nursing competence. However, it may vary depending on the support, culture, and atmosphere of the organization. In the early stages of clinical practice, nurses’ competence increases rapidly [23]. The knowledge, habits, and attitudes acquired during this period become the foundation for nursing practice throughout life. However, this period has a high turnover rate [6]. Therefore, organizing programs for motivation, knowledge, and attitude as experts for nurses in the early days of employment will not only increase their competence and capabilities but also reduce the turnover, thus improving the quality of nursing. In addition, nurses’ competence is directly related to patients’ safety; therefore, continuous competence development is needed [24,25]. In this sense, the HNCS can be used as a suitable scale for measuring nurses’ competence. However, as holistic nursing competence differs according to clinical experience and order of year, it is necessary to standardize scores according to the order of year. Similar to nurses’ evaluation on the clinical career ladder system in a hospital, holistic nursing competence must be continuously evaluated and developed. As the clinical career ladder system requires revision with changes over time, in the future, it should evaluate whether nurses can handle technical systems, such as AI and Internet of Things, which have evolved in the Fourth Industrial Revolution. Nonetheless, holistic nursing competence is universal as it is essential for nurses even with the passage of time and a changing environment.

As its validity and reliability were verified, the scale is believed to be helpful in measuring and developing the holistic nursing competence of clinical nurses. Nevertheless, this study has limitations. First, participants were selected from four general hospitals for convenience. Thus, the overall holistic nursing competence of all clinical nurses was not represented. Data in future studies should be collected through stratified random sampling. Second, this scale measures holistic nursing care competence through self-assessment. Self-evaluation justifies self-defined preferences and activities [1]. Thus, when measuring behavior, self-assessment and evaluation of others should be considered together using standard tools. Third, a test–retest method was not applied to identify the reliability of the HNCS. Last, the results should be verified for consistency across various hospital settings. 

## 5. Conclusions

The need for holistic nursing competence is increasing because of the absence of family care due to a reduction in the number of family members and an increase in isolation care due to rising infectious diseases in Korea. This study verified the Korean version of the HNCS. In the future, this scale should be used to measure, and subsequently develop, nurses’ competency. Consequently, nursing care quality will continue to improve. 

## Figures and Tables

**Figure 1 ijerph-19-07244-f001:**
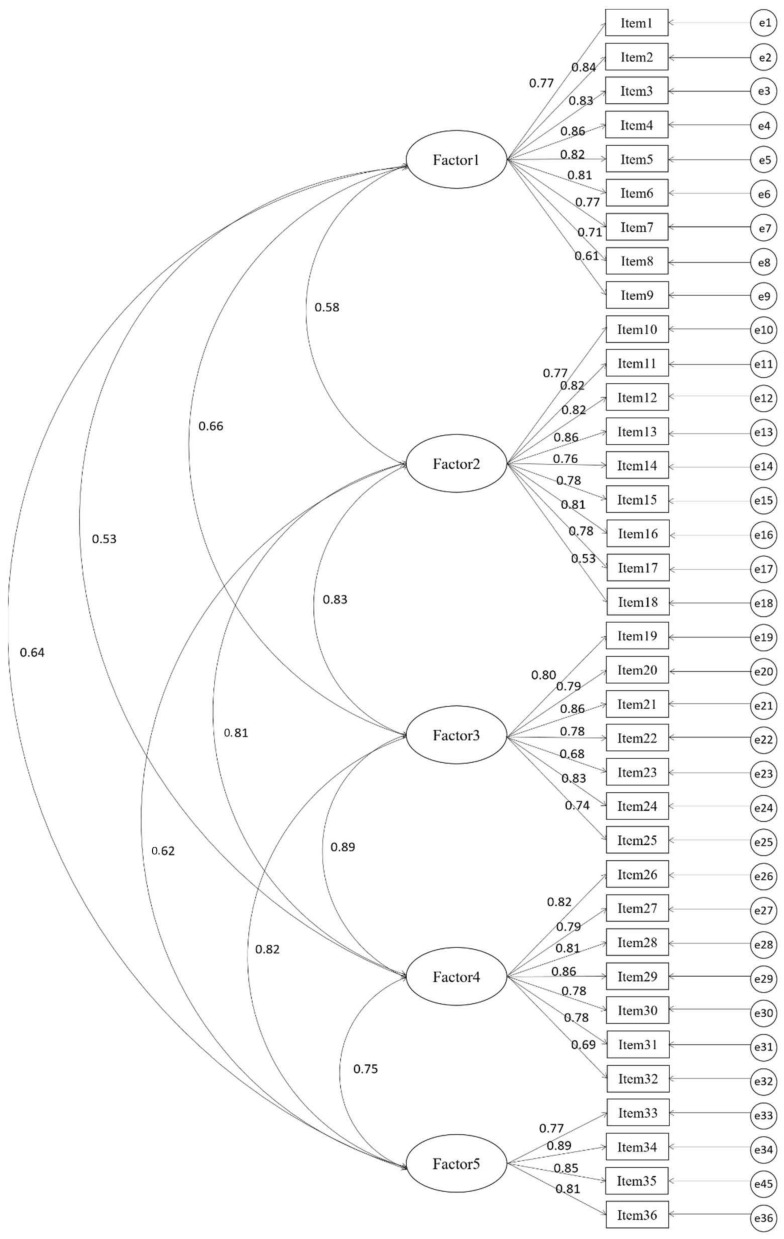
Measurement model of the Korean version of Holistic Nursing Competence Scale (HNCS).

**Table 1 ijerph-19-07244-t001:** General characteristics of participants and score according to factor (N = 251).

Characteristics	Categories	N (%) or M ± SD
Age (years)	≤29	132 (52.6)
	30–39	101 (40.2)
	40–49	14 (5.6)
	≥50	4 (1.6)
	Range 23–53	30.67 ± 5.48
Gender	Male	23 (9.2)
	Female	228 (90.8)
Marital status	Single	184 (73.3)
	Married	67 (26.7)
Religion	No	177 (70.5)
	Yes	74 (29.5)
Education	Diploma	27 (10.8)
	Bachelor	212 (84.5)
	≥Master	12 (4.8)
Total clinical career (years)	1–<5	124 (49.4)
	5–<10	68 (27.1)
	10–<15	36 (14.3)
	15–<20	15 (6.0)
	≥20	8 (3.2)
	Range 1–31	6.51 ± 5.68
Position	Staff nurse	244 (97.2)
	Above charge nurse	7 (2.8)
Work unit	Intensive care unit	95 (37.8)
	General unit	156 (62.2)
Factor 1: Staff education and management		4.116 ± 1.064
Factor 2: Ethically oriented practice		5.311 ± 0.875
Factor 3: General attitude		5.059 ± 0.920
Factor 4: Nursing care in team		5.274 ± 0.911
Factor 5: Professional development		4.935 ± 0.918

M = mean; SD = standard deviation.

**Table 2 ijerph-19-07244-t002:** Item analysis of HNCS (N = 251).

Dimension	Item	M ± SD	Skewness	Kurtosis
Staff education and management	Item1	4.04 ± 1.29	0.25	−0.26
	Item2	4.37 ± 1.33	−0.05	−0.58
	Item3	4.08 ± 1.33	0.03	−0.34
	Item4	4.11 ± 1.29	0.04	−0.23
	Item5	4.22 ± 1.25	−0.28	−0.18
	Item6	4.04 ± 1.32	0.08	−0.54
	Item7	4.21 ± 1.30	−0.01	−0.31
	Item8	4.05 ± 1.41	0.00	−0.64
	Item9	3.93 ± 1.35	−0.04	−0.46
	Total	4.12 ± 1.06		
Ethically oriented practice	Item10	5.09 ± 1.10	−0.12	−0.30
	Item11	5.25 ± 1.09	−0.20	−0.55
	Item12	5.24 ± 1.10	−0.16	−0.59
	Item13	5.27 ± 1.11	−0.27	−0.53
	Item14	5.09 ± 1.07	−0.27	−0.17
	Item15	5.33 ± 1.11	−0.36	−0.05
	Item16	5.35 ± 1.08	−0.26	−0.48
	Item17	5.69 ± 1.09	−0.63	0.03
	Item18	5.48 ± 1.14	−0.35	−0.58
	Total	5.31 ± 0.88		
General attitude	Item19	5.13 ± 1.05	−0.08	−0.56
	Item20	5.14 ± 1.08	−0.27	−0.25
	Item21	5.07 ± 1.07	−0.18	−0.46
	Item22	4.97 ± 1.09	−0.23	−0.23
	Item23	4.86 ± 1.24	−0.17	−0.53
	Item24	5.0 ± 1.19	−0.35	−0.40
	Item25	5.17 ± 1.19	−0.39	−0.33
	Total	5.06 ± 0.92		
Nursing care in a team	Item26	5.32 ± 1.06	−0.31	−0.26
	Item27	5.32 ± 1.10	−0.43	−0.07
	Item28	5.20 ± 1.15	−0.52	−0.01
	Item29	5.20 ± 1.14	−0.32	−0.49
	Item30	5.24 ± 1.11	−0.40	−0.35
	Item31	5.47 ± 1.02	−0.39	−0.46
	Item32	5.16 ± 1.18	−0.40	−0.27
	Total	5.27 ± 0.91		
Professional development	Item33	4.83 ± 1.05	−0.10	−0.38
	Item34	4.96 ± 1.01	−0.47	0.50
	Item35	4.86 ± 1.09	−0.25	−0.24
	Item36	5.09 ± 1.04	−0.21	−0.34
	Total	4.94 ± 0.92		

M = mean; SD = standard deviation.

**Table 3 ijerph-19-07244-t003:** Summary of fit indices from confirmatory factor analysis (N = 251).

Variables	CMIN/df	GFI	RMR	RMSEA	CFI	TLI	IFI
Evaluation criteria	≤3	≥0.90	≤0.05–0.08	≤0.05–0.08	≥0.90	≥0.90	≥0.90
HNCS	2.083	0.784	0.066	0.066	0.913	0.906	0.913

CMIN = chi-square minimum; df = degree of freedom; GFI = goodness-of-fit index; RMR = root mean square residual; RMSEA = root mean square error of approximation; CFI = comparative fit index; TLI = Tucker–Lewis index; IFI = incremental fit index.

**Table 4 ijerph-19-07244-t004:** Convergent validity.

Dimension	Item	Nonstandardized Estimate	SE	Critical Ratio	Standardized Estimate	AVE	CR ^1^
Staff education and management	Item9	1.00			0.61	0.829	0.890
	Item8	1.23	0.13	9.37	0.71		
	Item7	1.22	0.12	9.85	0.77		
	Item6	1.30	0.13	10.24	0.81		
	Item5	1.26	0.12	10.37	0.82		
	Item4	1.35	0.13	10.66	0.86		
	Item3	1.35	0.13	10.44	0.83		
	Item2	1.37	0.13	10.52	0.84		
	Item1	1.22	0.12	9.92	0.77		
Ethically-oriented practice	Item10	1.00			0.77	0.918	0.917
	Item11	1.05	0.07	14.07	0.82		
	Item12	1.06	0.08	14.20	0.82		
	Item13	1.13	0.08	15.08	0.86		
	Item14	0.96	0.07	12.86	0.76		
	Item15	1.03	0.08	13.41	0.78		
	Item16	1.03	0.07	13.86	0.81		
	Item17	1.01	0.08	13.38	0.78		
	Item18	0.72	0.08	8.54	0.53		
General attitude	Item25	1.00			0.74	0.872	0.895
	Item24	1.12	0.08	13.64	0.83		
	Item23	0.94	0.09	10.84	0.68		
	Item22	0.97	0.08	12.74	0.78		
	Item21	1.034	0.07	14.13	0.86		
	Item20	0.96	0.08	12.93	0.79		
	Item19	0.95	0.07	13.05	0.80		
Nursing care in a team	Item26	1.00			0.82	0.922	0.904
	Item27	1.00	0.07	14.52	0.79		
	Item28	1.08	0.07	14.98	0.81		
	Item29	1.14	0.07	16.47	0.86		
	Item30	1.00	0.07	14.15	0.78		
	Item31	0.93	0.07	14.30	0.78		
	Item32	0.94	0.09	12.12	0.69		
Professional development	Item36	1.00			0.81	0.883	0.904
	Item35	1.10	0.07	15.72	0.85		
	Item34	1.07	0.06	16.78	0.89		
	Item33	0.96	0.07	13.71	0.77		

SE = standard error; AVE = average variance extracted; CR ^1^ = construct reliability.

**Table 5 ijerph-19-07244-t005:** Discriminant validity (AVE > ρ^2^).

Variables	Correlation Coefficient (ρ^2^)				
	Factor 1	Factor 2	Factor 3	Factor 4	Factor 5
Factor 1	0.829 ^†^				
Factor 2	0.585(0.342) *	0.918 ^†^			
Factor 3	0.657(0.432) *	0.834(0.696) *	0.872 ^†^		
Factor 4	0.534(0.285) *	0.808(0.653) *	0.886(0.785) *	0.922 ^†^	
Factor 5	0.635(0.403) *	0.616(0.379) *	0.823(0.677) *	0.751(0.564) *	0.883 ^†^

AVE = average variance extracted; ^†^ = the value of the diagonal is AVE; * = *p* < 0.001.

**Table 6 ijerph-19-07244-t006:** Concurrent validity.

	Task Performance Evaluation Instrument Correlation (r)					Holistic Nursing Competence Scale Correlation (r)				
	a	b	c	d	e	1	2	3	4	5
a. Knowledge	1									
b. Attitude	0.783 **	1								
c. Ethics	0.679 **	0.744 **	1							
d. Performance	0.589 **	0.634 **	0.685 **	1						
e. Sum	0.873 **	0.930 **	0.875 **	0.813 **	1					
1. Factor 1	0.474 **	0.541 **	0.514 **	0.297 **	0.531 **	1				
2. Factor 2	0.457 **	0.534 **	0.527 **	0.454 **	0.565 **	0.551 **	1			
3. Factor 3	0.569 **	0.621 **	0.622 **	0.533 **	0.671 **	0.623 **	0.778 **	1		
4. Factor 4	0.561 **	0.653 **	0.580 **	0.542 **	0.674 **	0.508 **	0.753 **	0.818 **	1	
5. Factor 5	0.566 **	0.619 **	0.622 **	0.510 **	0.664 **	0.590 **	0.571 **	0.752 **	0.697 **	1
6. Sum	0.604 **	0.686 **	0.659 **	0.525 **	0.712 **	0.807 **	0.867 **	0.917 **	0.865 **	0.799 **

** *p* < 0.01.

## Data Availability

Not applicable.

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
