# Peer review of "Validity and Reliability of the Korean Version of the Holistic Nursing Competence Scale"

_ijerph, 2022, doi:10.3390/ijerph19127244_

Round 1

Reviewer 1 Report

Dear Authors many thanks for the changes made. In my opinion remain only minor changes. Please introduce in the background or in the abstract something related to the other used instrument 'Task Performance Evaluation Instrument for Clinical Nurses', which you never mention by referring only to the holist nursing scale.

Author Response

Thank you for your positive feedback. Per your comment, I have included, in the Abstract, the other instrument (Task Performance Evaluation Instrument for Clinical Nurses) that was used in this study. Additionally, I have also described this tool in the Introduction section (on page 2).

Reviewer 2 Report

Thank you very much for this revision of your manuscript. I think the changes you have made are helpful to the reader to follow what you have done. The manuscript presents a comprehensive psychometric evaluation of an important topic. 

I do not have any further comments to make about the manuscript.

Author Response

Thank you for your positive feedback. 

This manuscript is a resubmission of an earlier submission. The following is a list of the peer review reports and author responses from that submission.

Round 1

Reviewer 1 Report

Dear Authors, thanks for the opportunity to review this paper.
Below are the comments and suggestions to review the manuscript.
The abstract section is relatively poor. More details on the adapted and tested instruments, i.e. the total dimensions or the overall items, are necessary. The total sample size and the type of included nurses are lacking. The reliability indices are also lacking.
The introduction needs to be revised entirely. It is not clear why a holistic nursing scale is necessary. The connection between holistic nursing and the quality of care is not clear. It lacks a brief presentation of the other existing scales on holistic nursing or similar and why the authors chose this scale instead of another one. The reasons for the necessity of using this scale are missing. Moreover, the authors stated to test the psychometric properties, while they spoke only of validity and reliability in the introduction.
Methods are the essential section of a psychometric paper. The authors affirmed using a methodological research design, but no guidelines or methodological references on validation or cross-cultural adaptation are cited. The only cited reference was to the original scale.
Participants and procedure. The sampling method is unclear as to the inclusion criteria: was the sampling purposeful? How were the inclusion and exclusion criteria chosen? How many facilities were evaluated?, and why nurses without direct patients care were excluded?
The criteria used to select the sample to conduct the factor analysis were unclear; were 5 or 10 nurses for items? Moreover, it is unclear what the drop-out process is; I suggest that authors use a flowchart to present the sampling process. Lastly, the included settings were not specified. Where was the data collection conducted? How long was the data collection?
Measurements: The authors affirmed to have tested the psychometric properties of the Nursing Holist Scale, which is also presented as the 'Task Performance Evaluation Instrument for Clinical Nurses'. It is necessary to explain why the authors included it. Nothing about the scaling of the instrument is clearly understood. Perhaps it would be more appropriate to describe the macro-dimensions first and then the sub-dimensions. Is the Cronbach and authors' result, or was it the Cronbach reported by Takase? Unclear.
Results. I do not understand why content and face validity were not conducted (or not clearly stated?), was an expert panel consulted for adaptation? Also, did the authors first conduct an Exploratory Factor Analysis to assess the consistency of the items in the questionnaire? Furthermore, if so, it is possible to have a specification on the original items and the skewness and kurtosis of each item? It is much more than just the presentation of the original tool without even mentioning the supplementary file. The confirmatory factor analysis is well done. Are the authors thought about a second-order CFA? 

Author Response

Thank you for reviewing our manuscript again for publication in the IJERPH. We would like to thank the reviewers for their encouragement and through review. The reviewers’ comments have been addressed one by one. The reviewer’s comments are in italics; our responses are in bold in a log of our response to comments below. Changes that we have made are red-colored text in the manuscript and table (Please see the manuscript attached separately).

  1. The abstract section is relatively poor. More details on the adapted and tested instruments, i.e. the total dimensions or the overall items, are necessary. The total sample size and the type of included nurses are lacking. The reliability indices are also lacking.

Our response: We changed it as below

This methodological study aimed to verify the validity and reliability of the Korean version of the Holistic Nursing Competence Scale (HNCS), which comprises 36 items and five dimensions. The English version of the HNCS was forward and backward translated and administered to 251 participants with more than a year of work experience in a general hospital. Data were analyzed using SPSS WIN 24.0, and AMOS program was used for confirmatory factor analysis. Reliability assessed using Cronbach’s α was .969. Convergent, discriminant, and criterion validity were good. Average variance extracted and construct reliability ranged from .845 to .932 and .980 to .987, respectively.

  1. The introduction needs to be revised entirely. It is not clear why a holistic nursing scale is necessary. The connection between holistic nursing and the quality of care is not clear. It lacks a brief presentation of the other existing scales on holistic nursing or similar and why the authors chose this scale instead of another one. The reasons for the necessity of using this scale are missing. Moreover, the authors stated to test the psychometric properties, while they spoke only of validity and reliability in the introduction.

Our response: We deleted QoL and changed it as below

"Nurses account for the largest proportion of the hospital workforce and are a major factor influencing patient outcomes such as complication, infection, and death [6]. However, due to nurse shortage and excessive workload, nurses are unable to provide holistic nursing [1]. The fourth industrial revolution enables ICT-based smart healthcare services, including robots and artificial intelligence (AI), which can provide medical record-based nursing to patients [7]. Thus, nurses are able to provide more patient-centered nursing care in Korea. However, in line with these changes, nurses need to provide patient-centered care, which focuses on communication, shared understanding, and improved quality of life [7]. Therefore, holistic nursing competence is essential for providing patient-centered care."

  1. Methods are the essential section of a psychometric paper. The authors affirmed using a methodological research design, but no guidelines or methodological references on validation or cross-cultural adaptation are cited. The only cited reference was to the original scale.
    Participants and procedure. The sampling method is unclear as to the inclusion criteria: was the sampling purposeful? How were the inclusion and exclusion criteria chosen? How many facilities were evaluated?, and why nurses without direct patients care were excluded?
    The criteria used to select the sample to conduct the factor analysis were unclear; were 5 or 10 nurses for items? Moreover, it is unclear what the drop-out process is; I suggest that authors use a flowchart to present the sampling process. Lastly, the included settings were not specified. Where was the data collection conducted? How long was the data collection?

Our response:

1) Reference for methodological and cross cultural adaptation is Waltz et. al(2016). Reference number is [16].

2) Data were collected 4 general hospitals. It was convenience extraction. It was difficult to survey from nurses at the hospital. Supervisors worried about nurses' stress and overwork due to COVID-19.

3) Sample size: Two criteria were presented, but one was deleted for clarity.

4) Reasons for excluding nurses under 1 year: They would have difficulties in job and adaptation.

5) Reasons for excluding nurses who are not directly care for patients: Takase and Teraoka excluded nurses who did not provide directly care. I think same as that. Nurses whose main duties were administration, explanation, and injection were excluded.

6) Hospital with 300 beds or less : It was a matter expression. We didn’t mean to intentionally exclude it. There were four hospitals that allowed the survey, and all of them were general hospitals with more than 300 beds. So the sentence was excluded.

7) We changed as below

"The study participants were nurses with more than a year of work experience in four general hospitals, and those who voluntarily participated in the study. New nurses with less than a year of work experience were excluded because it was assumed that they face more difficulties in adapting to and performing their job [11, 12]. Moreover, nurses working in departments that did not provide direct nursing care, such as outpatient clinics, department of infection control and management, department of healthcare quality management, and administrative departments, were excluded. This was so because the test items explored relationships, communication with patients, and education for patients and caregivers. The recommended sample size for the confirmatory factor analysis (CFA) using the AMOS program was 200–400 [13]. The data of 251 participants were used for analysis."

  1. Measurements: The authors affirmed to have tested the psychometric properties of the Nursing Holist Scale, which is also presented as the 'Task Performance Evaluation Instrument for Clinical Nurses'. It is necessary to explain why the authors included it. Nothing about the scaling of the instrument is clearly understood. Perhaps it would be more appropriate to describe the macro-dimensions first and then the sub-dimensions. Is the Cronbach and authors' result, or was it the Cronbach reported by Takase? Unclear.

Our response:

1) We added the reason for using the Task Performance Evaluation Instrument for Clinical Nurses.

"Task Performance Evaluation Instrument for Clinical Nurses is a tool for evaluating the work performance ability of clinical nurses [14]. Most of evaluation methods used in hospitals are methods in which supervising nurses evaluated staff nurses [14]. This tool was designed to address the problem of the one--sided evaluation method by superiors. In other words, it simultaneously considers nurses’ self-assessment and supervisors’ evaluation. It comprises four dimensions and 35 items, which are knowledge (eight items), attitude (thirteen items), performance ability (seven items), and ethics (seven items). According to several researchers, this tool contains the necessary components to assess holistic nursing care and nurses’ competency [15]. This questionnaire employs a 5-point Likert scale ranging from one to five, and its score ranges from 35–175. The higher the score, the better the task performance. The reliability of the questionnaire at the time of development was .918 whereas Cronbach’s α was .959 in this study."

2) Cronbach’s α was Takase. This parts has been corrected.

  1. Results. I do not understand why content and face validity were not conducted (or not clearly stated?), was an expert panel consulted for adaptation? Also, did the authors first conduct an Exploratory Factor Analysis to assess the consistency of the items in the questionnaire? Furthermore, if so, it is possible to have a specification on the original items and the skewness and kurtosis of each item? It is much more than just the presentation of the original tool without even mentioning the supplementary file. The confirmatory factor analysis is well done. Are the authors thought about a second-order CFA?

Our response:

1) We corrected '2.4 procedure' and added content validity of the results

2) Because skewness and Kurtosis were not added in Takase's study, our results were not included. We added it on table2.

3) We thought about a second-order CFA. We plan to further study HNCS.

Reviewer 2 Report

Thank you for the opportunity to review this manuscript. I understand it is a psychometric evaluation and this is an important step in measurement tool development. It always should come before evaluating the implementation and use of a tool in clinical practice. 

Holistic nursing care competence is an important concept and being able to measure it is important. Hence this work is important. 

In reading the manuscript I found it is in need of a thorough edit for English and scientific language. There are many places where the English presentation is confusing and the reader is left to make assumptions about meaning. In addition, there are details that would be helpful to  reader to understand the work if they could be incorporated into the text. 

1) In the abstract: both sentence 2 and 3 are confusing as worded.

2) Line 40: the phrase 'developing insufficient competencies' does not make sense here and sounds quite wrong.

3) Exclusion criteria - why are you excluding hospitals with <300 beds? Holistic Nursing Competence should be a consistent performance by all nurses regardless of where they work

4) In thinking about the response items on the scale please clarify the self report nature of the questionnaire (nurses are reporting on their own behavior) and this limitation; also discuss the difference in asking to report how often a nurse performs a competency and how comfortable she/he feels in performing a competency; which did this Korean scale do?   

5) When introducing the HNSC it would be important to describe more about how it was developed and the nature of the items. How do you know it actually measures competency? Is it also a self report questionnaire? Was it every tested against actual competency versus self reported copetency?

6) In introducing the Task Performance evaluation tool, how was it developed? Do the items on it match the HNSC? in 2.3.2 you ought to describe how you plan to use it and why?

7) you say you want to test content, construct, concurrent validity and reliability; but then later you report on criterion and discriminant validity. A reader can be confused about this. Be consistent or provide definitions of hte type of validity you are testing. 

8) It is also important to describe the hypotheses that are driving your analysis rather than just reporting results. 

9) I am surprised that you seemed to combine paper and on-line approaches. These are really two different approaches for completion and could well have different reliabilities. 

10) in reporting on content validity, you say you assessed appropriateness of wording. This is one aspect of content validity. How do you know the tool really assessed competency for holistic nursing practice. Did you have a framework against which to assess? 

11) In the discussion, the opening paragraph really belongs back with the introduction of the tool itself. (See earlier comment). And the second paragraph can be summarized and not report so many of the actual numbers.

12) At line 288 you say you were verifying the usefulness through this psychometric work. I would disagree, The usefulness would require a very different piece of work. 

13) line 295 - I challenge your assumption that holistic competence varies. It should not - rather it ought to be consistent across settings.

14) 'Absence of family care' ought to be explained more fully for the reader.

Author Response

Manuscript ID: ijerph-1695928 entitled “Exploring the psychometric properties of the Holistic Nursing Competence scale

Thank you for reviewing our manuscript again for publication in the IJERPH. We would like to thank the reviewers for their encouragement and through review. The reviewers’ comments have been addressed one by one. The reviewer’s comments are in italics; our responses are in bold in a log of our response to comments below. Changes that we have made are red-colored text in the manuscript and table (Please see the manuscript attached separately).

  1. 1. In the abstract: both sentence 2 and 3 are confusing as worded.

Our response: We revised the abstract as a whole.

  1. Line 40: the phrase 'developing insufficient competencies' does not make sense here and sounds quite wrong.

Our response: That was an unclear sentence. We deleted it.

  1. Exclusion criteria - why are you excluding hospitals with <300 beds? Holistic Nursing Competence should be a consistent performance by all nurses regardless of where they work

Our response: That sentence is wrong. Four hospitals were convenience sampled. Four hospitals were inadvertently general hospitals with more than 300beds.

  1. 4. In thinking about the response items on the scale please clarify the self report nature of the questionnaire (nurses are reporting on their own behavior) and this limitation; also discuss the difference in asking to report how often a nurse performs a competency and how comfortable she/he feels in performing a competency; which did this Korean scale do?

Our response: We added it on the limitations.  The Korean scale asked for frequency. We discussed about asking for frequency and competence.

  1. When introducing the HNSC it would be important to describe more about how it was developed and the nature of the items. How do you know it actually measures competency? Is it also a self report questionnaire? Was it every tested against actual competency versus self reported copetency?

Our response: We described it as a limitation of this study.

  1. In introducing the Task Performance evaluation tool, how was it developed? Do the items on it match the HNSC? in 2.3.2 you ought to describe how you plan to use it and why?

Our response: The dimensions and items of task performance evaluation tool match the HNCS. We added a description.

7 you say you want to test content, construct, concurrent validity and reliability; but then later you report on criterion and discriminant validity. A reader can be confused about this. Be consistent or provide definitions of hte type of validity you are testing.

Our response: We described them in order (content, convergent, discriminant, criterion validity and reliability).

  1. It is also important to describe the hypotheses that are driving your analysis rather than just reporting results.

Our response: We added it.

  1. I am surprised that you seemed to combine paper and on-line approaches. These are really two different approaches for completion and could well have different reliabilities.

Our response: We planned an online survey, but one of four hospitals wanted paper survey. It will be verify it through a follow-up study.

10 in reporting on content validity, you say you assessed appropriateness of wording. This is one aspect of content validity. How do you know the tool really assessed competency for holistic nursing practice. Did you have a framework against which to assess?

Our response: We performed CVI but did not describe it. It was added.

  1. In the discussion, the opening paragraph really belongs back with the introduction of the tool itself. (See earlier comment). And the second paragraph can be summarized and not report so many of the actual numbers.

Our response: Some of the contents of a tool were deleted from the opening paragraph. Actual numbers are likely to be needed to help readers understand.  

  1. At line 288 you say you were verifying the usefulness through this psychometric work. I would disagree, The usefulness would require a very different piece of work.

Our response: It was deleted.

  1. line 295 - I challenge your assumption that holistic competence varies. It should not - rather it ought to be consistent across settings.

Our response: We are same opinion as you. The sentence is incorrect. It was deleted.

  1. 'Absence of family care' ought to be explained more fully for the reader.

Our response: ‘Absence of family care’ led to the creation of an integrated nursing care service. We added contents related to family care.   

“In Korea, a family member is allowed to attend to an inpatient in a hospital. However, due to increased emphasis on individual quality of life and other social changes, there is a decline in family care being provided. For inpatient care, many people hire caregivers, which can be costly. To help individuals with financial difficulties, the Korean government introduced the integrated nursing care service in 2013 and began to establish hospitals without family caregivers [8]. All care is provided by nurses without a family caregivers in the integrated nursing care service. In other word, this means that not all nursing care for the patient is done in the presence of the caregiver, but by the nurse themselves, so that the family care for patients is reduced.”

Reviewer 3 Report

I have several observations on the manuscript and hope my suggestion can help to improve the content and presentation.

literature review:

the research problem is notc lear enough. why did the authors want to translate the questionnaire? Is it currently not having a Korean nursing competence scale? what is the uniqueness of this scale that makes the team selected this scale but not other for the translation. i think it is important to have a convincing research problem. 

study design/data analysis:

  • content validity: more description is needed on the review panel composition, items included in the review and how to calculation the CVI.
  • for reliability, only internal consistency was reported. usually we did test-retest as well to ensure stability of the translated questionnaire. see if the authors can also include this result in the manuscript or state it as a limitation in the discussion if not done.

Result: clear description of the findings. 

Discussion and conclusion: the discussion has addressed the study topic and explained the findings. Limitations or recommendations for further research to ensure the scale meeting higher psychometric property will be better. 

Author Response

Manuscript ID: ijerph-1695928 entitled “Exploring the psychometric properties of the Holistic Nursing Competence scale

Thank you for reviewing our manuscript again for publication in the IJERPH. We would like to thank the reviewers for their encouragement and through review. The reviewers’ comments have been addressed one by one. The reviewer’s comments are in italics; our responses are in bold in a log of our response to comments below. Changes that we have made are red-colored text in the manuscript and table (Please see the manuscript attached separately).

  1. 1. literature review: the research problem is not clear enough. why did the authors want to translate the questionnaire? Is it currently not having a Korean nursing competence scale? what is the uniqueness of this scale that makes the team selected this scale but not other for the translation. i think it is important to have a convincing research problem.

Our response: We revised the introduction.

  1. 2. study design/data analysis:
  • content validity: more description is needed on the review panel composition, items included in the review and how to calculation the CVI.
  • for reliability, only internal consistency was reported. usually we did test-retest as well to ensure stability of the translated questionnaire. see if the authors can also include this result in the manuscript or state it as a limitation in the discussion if not done.

Our response: The review panel composition is 10 people. We performed CVI but did not describe it. It was added. Test-retest was not performed. We stated it as a limitation in the discussion.

  1. 3. Result: clear description of the findings.

Our response: We tried to describe the results themselves.

  1. 4. Discussion and conclusion: the discussion has addressed the study topic and explained the findings. Limitations or recommendations for further research to ensure the scale meeting higher psychometric property will be better.

Our response: It was revised (from 299 line).  

Round 2

Reviewer 1 Report

Dear authors and thanks again for the opportunity to review this paper.

Unfortunately, I believe that the various changes made are not sufficient for a publication. The abstract has been clearly modified as well as the various additions made in the methods section, but it is lasck clarity on the motivations which support the validation of the scale. The introduction should be more convincing, the added value for nursing to validate this scale continues to be unclear. Methods are confusing (some information was included in the discussions i.e. the re-test performed for validation) and the inclusion criteria continue to be unclear as well as the methodological steps taken. As previously stated if a validation manuscript is poor in methodology, it is a major issue.